# Enhancing reliability in electrical grids: A hybrid machine learning approach for electrical faults classification

Momotaz Begum[1], Ariful Islam Shiplu[1], Mehedi Hasan Shuvo[1], Fahmid Al Farid[2], Sumaiya Ismat Jerin[3], Jia Uddin[4]*, Hezerul bin Abdul Karim[2]*

**1** Department of Computer Science and Engineering, Dhaka University of Engineering & Technology (DUET), Gazipur, Bangladesh, **2** Centre for Image and Vision Computing (CIVC), COE for Artificial Intelligence, Faculty of Artificial Intelligence and Engineering (FAIE), Multimedia University, Cyberjaya, Selangor, Malaysia, **3** Department of Computer Science and Engineering, Green University of Bangladesh, Kanchan, Bangladesh, **4** AI and Big Data Department, Endicott College, Woosong University, Daejeon, South Korea

* jia.uddin@wsu.ac.kr (JU); hezerul@mmu.edu.my (HAK)

## Abstract

Transmission lines are vital components of electrical grids, ensuring the efficient transfer of electricity from power plants to consumers over extensive geographical areas. These lines are constructed with careful consideration of factors such as conductor materials, insulation levels, current ratings, and voltage ratings to maintain reliable and safe electricity delivery. However, various types of faults can occur in transmission lines, posing significant challenges, often leading to outages, equipment damage, and reduced system reliability. Accurate and fast fault classification is therefore a pressing requirement in modern smart grids, where proactive maintenance and resilience are critical. This research addresses the critical need for an efficient electric fault classification model. A comprehensive investigation is conducted, employing a variety of machine learning (ML) algorithms, including Decision Tree (DT), Random Forests (RF), Naive Bayes (NB), K-Nearest Neighbors (KNN), Support Vector Machine (SVM), and, AdaBoost, for fault classification. Additionally, fundamental ensemble techniques such as Hard-Voting, Soft-Voting, Stacking, and Blending are incorporated with five hybrid ML models (each constructed by combining various ML algorithms) to enhance fault classification performance and the reliability of transmission lines. Also, this research proposes a hybrid ML model, specifically *(RF + DT + Stacking)*, to classify transmission line data. The main contribution of this work is an application-oriented evaluation of classical and ensemble machine learning models for electrical fault classification, with an emphasis on benchmarking performance, model interpretability, and computational efficiency. This study demonstrates that a carefully configured hybrid ensemble *(RF + DT + Stacking)* can provide a practical and lightweight alternative to deep learning-based methods in grid fault monitoring scenarios. The dataset used encompasses various attributes affecting line

**Data availability statement:** The data that supported the findings of this study are available at the following link: https://www.kaggle.com/datasets/esathyaprakash/electrical-fault-detection-and-classification.

**Funding:** This research was funded by Multimedia University, Cyberjaya, Selangor, Malaysia (Grant Number: PostDoc (MMUI/240029)). The funders had no role in study design, data collection and analysis, decision to publish, or preparation of the manuscript.

**Competing interests:** No authors have competing interests.

performance, making accurate classification critical for proactive issue detection, optimized maintenance scheduling, and uninterrupted energy supply. Our hybrid model achieves high-performance metrics, including an accuracy of 93.64%, precision of 93.65%, recall of 93.64%, and F1 score of 93.64%, underscoring its effectiveness in enhancing decision-making processes and operational efficiency within electrical transmission networks.

## 1 Introduction

Transmission lines are high-voltage power lines designed to carry electricity from power plants to substations over long distances with minimal losses. They facilitate the delivery of power generated at remote plants to end-users, ensuring reliable electricity supply in various sectors of society. The design and operation of transmission lines encompass several key considerations, including conductor material, insulation levels, and voltage ratings, which optimize energy transmission while prioritizing reliability and safety. Their versatility allows them to carry both alternating current (AC) and direct current (DC), further enhancing their importance in maintaining a stable power supply [1].

Key factors in the design of transmission lines include length, voltage, and current capacity. A careful balance among these parameters is essential to meet demand while minimizing energy losses [2]. Additionally, the condition of transmission lines, particularly insulation levels, is crucial for preventing outages and ensuring public safety [3]. Various types of transmission lines, such as overhead lines, underground lines, and submarine cables, each possess unique advantages and disadvantages suited to specific applications, highlighting the necessity of tailoring solutions to operational environments [4].

Transmission lines operate under diverse environmental and physical conditions that can significantly impact their performance and lifespan. Factors such as temperature fluctuations, weather conditions, and physical wear necessitate proper maintenance and regular inspections to identify and mitigate potential issues that could lead to failures [5]. Advanced monitoring systems and diagnostic tools are increasingly employed to assess the condition of transmission lines, facilitating the early detection of anomalies that may indicate underlying problems [6].

The advent of smart grid technologies offers promising opportunities for enhancing the management and operation of transmission lines. Smart grids leverage advanced sensors, communication networks, and data analytics to enable real-time monitoring and control of electrical infrastructure, allowing for improved load balancing, fault detection, and automatic grid reconfiguration in response to changing conditions [7]. Integrating smart grid technologies with transmission line infrastructure can significantly enhance reliability, resilience, and efficiency in electricity delivery [8].

ML has emerged as a transformative technology across various fields, enabling the development of models that can learn from data and make predictions without explicit programming. In the realm of electrical engineering, machine learning is increasingly utilized to analyze and predict the performance and condition of transmission lines, thus improving their reliability and efficiency [9]. These algorithms can process vast amounts of data to identify patterns, detect anomalies, and make accurate predictions, enabling proactive issue resolution and optimized maintenance schedules [10]. Ensemble methods were later explored to improve robustness and class-wise reliability without incurring the heavy training cost of deep neural network [11,12]. In parallel, the power-systems community investigated feature engineering from phase voltages/currents, sequence components, symmetrical components, and time–frequency transforms to increase separability across single-line-to-ground, line-to-line, and three-phase faults [13]. Robust control and estimation techniques have been widely applied in motor drives and power systems, such as H controllers with MRAS-based estimators [14,15] and adaptive model predictive control (AMPC) with online parameter estimation [16], to enhance dynamic performance and reliability. In the energy sector, machine learning and deep learning models are increasingly explored for tasks like supercapacitor performance prediction [17] and solar power forecasting [18].These studies show the potential of data-driven methods in renewable energy applications. Recent studies emphasize MPPT algorithms for PV systems [19], power quality in wind energy with DFIGs [20], optimization of PV and DSTATCOM placement [21], fault detection in PV modules via imaging [22], and hybrid power systems with demand response strategies [23], showcasing the breadth of advanced control and optimization in renewable energy. In addition, research has highlighted the role of ML/DL techniques in wind power prediction [24], integration challenges of renewable energy sources in modern grids [25], LVRT enhancement for wind farms using DFIG protection schemes [26], hybrid energy system optimization with novel metaheuristics [27], and fault detection in solar panels using ML classifiers [28], further reinforcing the transformative impact of intelligent methods in renewable energy systems.

Unlike general ML classification problems, electrical fault classification involves high-frequency transient behavior in voltages and currents. Fault events cause abrupt changes in current magnitudes (Ia, Ib, Ic) and subtle variations in voltage signals. These patterns are nonlinear and system-dependent, governed by power-system physics, line impedance, fault resistance, and sequence component behavior. The selected ML models are therefore applied to system-generated signals representing real electrical phenomena rather than generic tabular data.

**Scope:** This research focuses on developing a hybrid ML approach to improve the reliability and performance of electrical transmission lines by accurately classifying electrical faults. Utilizing a comprehensive dataset with attributes related to line performance, the study explores and evaluates several ML algorithms, including DT, RF, NB, KNN, SVM, and AdaBoost. It further integrates ensemble techniques like Hard-Voting, Soft-Voting, Stacking, and Blending, emphasizing the hybrid model *(RF + DT + Stacking)*. The scope includes enhancing fault classification accuracy, optimizing maintenance processes, and minimizing energy supply disruptions, ultimately contributing to more reliable and efficient grid operations.

The following are some of our study's contributions:

- Demonstrated that a hybrid ensemble configuration *(RF + DT + Stacking)* provides a competitive balance between classification performance, interpretability, and computational efficiency for transmission line fault classification. Using RF and DT as base learners within a stacking framework, the model achieves strong performance, with an accuracy of 93.64%, precision of 93.65%, recall of 93.64%, and an F1 score of 93.64%.
- Demonstrated the effectiveness of the hybrid model in accurately predicting the performance and condition of transmission lines, providing a robust and reliable framework for classification that can aid in early issue detection and optimized maintenance scheduling.
- Enhanced decision-making processes in electrical energy transmission by offering a data-driven approach that ensures reliability and safety, contributing to the field of electrical engineering by integrating multiple machine learning

techniques to improve classification accuracy and robustness. Showcased the practical applications and benefits of ML in optimizing power distribution systems, thereby enhancing their efficiency and reliability.

This paper investigates the following key research questions:

- **RQ1:** *What machine learning algorithms have been utilized for electrical fault classification, and what advantages do they offer in terms of accuracy, efficiency, and real-time fault classification?*
- **RQ2:** *How does an application-oriented hybrid ensemble configuration improve the balance between accuracy, interpretability, and computational efficiency in electrical fault classification?*
- **RQ3:** *Which hybrid machine learning model can outperform previous research and improve the accuracy of electrical fault classification?*

The organization of this paper is as follows: Sect 2 presents the Related Work, discussing previous research and methodologies relevant to electrical fault classification and machine learning. Sect 3 covers the Background Theory, providing an overview of this study's key concepts, algorithms, and techniques. Sect 4 describes the Proposed Approach, detailing the hybrid machine learning model developed for fault classification. Experiments and Results are presented in Sect 5, where we analyze the performance metrics of our model. Sect 6 engages in a thorough discussion of the results. Sect 7 concludes the paper with a summary of key findings in the Conclusion, while Sect 8 outlines potential Future Work, highlighting areas for further research and improvement.

## 2 Related work

ML techniques have gained significant attention in electrical engineering, particularly for the classification, and monitoring of transmission lines. Traditional methods, such as supervised learning, have been effective in various applications [29–31]; however, they often face challenges when handling large-scale datasets and achieving high classification accuracy. Ensemble learning techniques have emerged as a promising solution to these challenges by combining multiple ML models to enhance predictive performance [32]. Among these, the Stacking Ensemble method has shown superior accuracy and robustness in power system fault diagnosis [11,33]. For instance, researchers have applied Stacking approaches to fault detection in transmission lines, demonstrating improved results compared to single-model methods.

Recent advancements in transmission line fault diagnosis have highlighted the integration of ML with real-time data analytics. Sun et al. [34] introduced an improved multiple SVM model optimized by a genetic algorithm, achieving an accuracy improvement of up to 11%. Their method was validated on an IEEE-30 node test system and real-world data, effectively addressing issues related to small sample sizes and generalization accuracy.

Yin et al. [35] developed a predictive decision support system using data mining techniques, correlating multi-source dynamic datasets with meteorological data to model transmission line disasters. This approach provided high-accuracy early warnings for potential failures, underscoring its relevance in regions like Bangladesh, where extreme weather poses significant risks to the power grid. Additionally, Tong et al. [36] proposed a novel transient fault detection and classification approach utilizing graph convolutional neural networks (GCN). By incorporating spatial information from sampling sequences and topology data, their method has shown exceptional performance in real-time fault detection, offering advantages in online transmission line protection.

Furthermore, Yu et al. [37] developed a diagnostic approach utilizing the Elgamal encryption algorithm, which not only improved data security but also achieved diagnostic accuracy exceeding 90%. Ma et al. [38] established a simulation model to identify both lightning and non-lightning strike faults, proposing criteria based on transient traveling wave current characteristics for intelligent fault diagnosis. Lahiri et al. [39] introduced a fault diagnosis method employing Decision Tree and Random Forest techniques, achieving 95% to 100% accuracy in identifying the location, type, and faulty phase of transmission lines across various scenarios.

Agarwal et al. [40] proposed a method for rapid fault identification in line commutated converter-based high voltage DC transmission lines, utilizing discrete Fourier transform analysis of DC current to enhance accuracy and reliability for timely trip commands to DC breakers. Additionally, the work presented by Hao et al. [41] introduced a faulted phase selection scheme that utilizes Multiscale Principal Entropy (MPE) values from fault transient voltage signals, combined with CS-SVM for high-accuracy fault detection, resilient to variations in fault location and other parameters. Lastly, the 10kV railway power transmission line simulation model developed by Yu et al. [42] employed a BP neural network to classify faults based on phase current differences, accurately determining fault types and locations.

Despite these developments, there remains a need for more comprehensive models that integrate various ML techniques to maximize their strengths and mitigate weaknesses. Our proposed hybrid model, which combines *(RF + DT + Stacking)* Ensemble techniques, seeks to address these gaps by providing a more robust and accurate classification framework for transmission line data. In Bangladesh, where the electrical grid faces challenges from extreme weather conditions, hybrid models can enhance transmission line monitoring significantly [43]. By improving the reliability and efficiency of these systems, our research aims to contribute to stable power system operations, reducing economic losses, and ensuring consistent electricity supply.

This work builds upon existing research while introducing a hybrid approach that integrates multiple ML techniques, thereby contributing to advancements in electrical grid management and enhancing the reliability and efficiency of transmission line monitoring. Table 1 represents the contributions and limitations of literature review papers.

## 3 Background theory

In this section, we outline the foundational concepts and theoretical underpinnings relevant to this research. We will focus on the mathematical formulations of the supervised ML algorithms and ensemble methods utilized in this study.

### 3.1 Decision Tree (DT)

Decision Tree (DT) algorithm is renowned for its ease of use and clarity, which makes it a popular choice for data analysis, mainly when dealing with large datasets [47]. Its ability to perform automatic feature selection is a crucial advantage,

**Table 1**. Contributions and limitations of different studies in the literature.

| Ref. | Contributions | Limitations |
|---|---|---|
| [41] | Presented a faulted phase selection scheme utilizing Multiscale Principal Entropy (MPE) values from fault transient voltage signals combined with CS-SVM for high-accuracy fault detection. The method identifies faults rapidly and is resilient to variations in fault location, transition resistance, and initial angle state. | Requires extensive training data for the CS-SVM model to achieve optimal performance; effectiveness in diverse real-world conditions needs validation. Computational complexity may challenge real-time implementation. |
| [42] | Developed a 10kV railway power transmission line simulation model using MATLAB/Simulink, employing a BP neural network to classify faults based on phase current differences and zero-sequence current features. It derives a fault location formula and accurately determines fault types and locations. | Effectiveness relies on the quality and quantity of training data for the BP neural network, potentially impacting classification accuracy. Further validation in real-world scenarios is necessary for robustness confirmation. |
| [44] | Introduced a method using multiwavelet packet entropies with RBF neural network for classifying 10 types of transmission line faults, achieving optimal performance with SA4 multiwavelet packet Tsallis singular entropy | Computational complexity may limit real-time detection. |
| [45] | Proposed a novel method for fault classification in single-circuit transmission lines using one-side voltage and current, leveraging a DT algorithm trained on odd harmonics (up to the 19th), delivering rapid fault classification in under a quarter cycle with high accuracy. | The approach may face challenges with real-time performance in large systems due to harmonic processing complexity and also requiring further validation in practical environments. |
| [46] | Introduced a KNN-based method for fault detection, classification, and location in the IEEE 14 Bus Test Network using MATLAB/SIMULINK, enhancing protection system efficiency. | struggle with scalability and performance in larger networks and requires optimization of KNN parameters for better accuracy in diverse conditions. |

as it highlights the most influential features for accurate predictions. The DT represents data through a tree-like model, making decisions hierarchically that reveal valuable patterns and decision processes. Understanding the DT algorithm requires familiarity with concepts like Entropy (E), Gini Index (GI), and Information Gain (IG). These concepts are defined as follows: Certainly! Let's update the equations with different variable names for clarity:

$$H(D) = -\sum_{j=1}^{K} p(j) \log_2 p(j) \tag{1}$$

$$G(D) = 1 - \sum_{j=1}^{m} (p_j)^2 \tag{2}$$

$$IG(D, A) = H(D) - \sum_{k \in \text{Values}(A)} \frac{|D_k|}{|D|} H(D_k) \tag{3}$$

In these equations, $H(D)$ represents the entropy of the dataset $D$, $G(D)$ denotes the Gini impurity at a node, and $IG(D, A)$ indicates the information gain from an attribute $A$ with respect to dataset $D$.

### 3.2 Random Forest (RF)

The Random Forest (RF) classifier is an ensemble technique that aggregates multiple decision trees, each trained on different subsets of the dataset. By averaging the predictions of these trees, it enhances overall predictive accuracy. Renowned for its resilience, the RF classifier effectively manages diverse data challenges, including categorical variables, imbalanced datasets, and missing values. This approach leverages the combined strength of various decision trees, making it a robust and reliable method for delivering accurate predictions, even in complex data situations.

### 3.3 K-Nearest Neighbors (KNN)

The K-nearest neighbor (KNN) algorithm is well-regarded for its simplicity, ease of interpretation, and effectiveness in classification tasks. It operates by storing all available instances and classifying new instances based on their similarity to these stored cases, typically using distance metrics. By examining the $k$ closest neighbors, it assigns a class label based on the most common label among these nearby points. This straightforward method makes it easy to understand and applies effectively across various domains, including pattern recognition, recommendation systems, and data mining. The similarity between instances can be measured using the following distance metrics.

The general distance metric is given by:

$$\delta_p(a, b) = \left( \sum_{j=1}^{d} |a_j - b_j|^p \right)^{1/p} \tag{4}$$

where $a$ and $b$ are two vectors. The Manhattan distance (also known as Taxicab distance) is expressed as:

$$\delta_M(u, v) = |u_1 - v_1| + |u_2 - v_2| \tag{5}$$

where $(u_1, v_1)$ and $(u_2, v_2)$ are two points in a plane. The Euclidean distance is given by:

$$\delta_E = \sqrt{(u_2 - u_1)^2 + (v_2 - v_1)^2} \tag{6}$$

where $(u_1, v_1)$ and $(u_2, v_2)$ are points in a two-dimensional space. The formula for making predictions with the KNN algorithm is:

$$\hat{y} = \text{mode}(y_j)_{j \in \text{KNN}(x)} \tag{7}$$

In this equation, $\hat{y}$ represents the predicted class for a new input $x$, $\text{KNN}(x)$ denotes the set of $K$ nearest neighbors to $x$, and $y_j$ is the class label associated with the $j$-th neighbor in the set. The function `mode` selects the most frequently occurring class label among the neighbors.

## 3.4 Naive Bayes (NB)

Bayes' theorem provides a straightforward way to quantify the probability of a true hypothesis based on specific evidence. It is a core concept in probability theory that facilitates the calculation of conditional probabilities. NB algorithm, a widely used method in ML, is built upon Bayes' theorem. Below are the reformulated equations for Bayes' theorem:

The product rule can be expressed as:

$$P(A \cap B) = P(A \mid B) \cdot P(B) \tag{8}$$

Since:

$$P(A \cap B) = P(B \cap A) \tag{9}$$

It follows that:

$$P(A \cap B) \cdot P(B) = P(B \mid A) \cdot P(A) \tag{10}$$

Bayes' theorem itself is given by:

$$P(A \mid B) = \frac{P(B \mid A) \cdot P(A)}{P(B)} \tag{11}$$

In this context, $A$ represents the hypothesis and $B$ denotes the evidence. $P(A \mid B)$ denotes the posterior probability of the hypothesis $A$ given the evidence $B$. $P(B \mid A)$ is the probability of observing the evidence $B$ given that the hypothesis $A$ is true. $P(A)$ and $P(B)$ are the prior probabilities of the hypothesis and evidence, respectively.

## 3.5 Support Vector Machine (SVM)

Support Vector Machine (SVM) is known for its simplicity and effectiveness as an ML algorithm, offering precise results with relatively low computational demands. It excels at linear and nonlinear classification tasks by maximizing the margin between data points and a decision boundary, typically represented as a hyperplane or line. SVM is also effective at preventing overfitting. The kernel trick also allows SVM to handle more complex, non-linear problems.

## 3.6 AdaBoost and gradient boosting

AdaBoost is an effective ML algorithm that combines multiple weak learners to improve overall prediction performance. AdaBoost enhances accuracy by integrating these relatively simple models, resulting in a more robust and reliable predictive system. In contrast, GB is another ML technique used for regression and classification tasks. When applied to regression trees, GB produces strong, resilient models capable of handling noisy or incomplete data. The final prediction in GB is determined using the following formula:

$$\mathcal{L}(h) = \sum_{i=1}^{m} \mathcal{L}(t_i, h(z_i)) \tag{12}$$

The loss function $\mathcal{L}(h)$ can be minimized with respect to $h$ using:

$$\hat{h}_0(z) = \arg\min_h \mathcal{L}(h) = \arg\min_h \sum_{i=1}^{m} \mathcal{L}(t_i, h(z_i)) \tag{13}$$

In these equations, $t_i$ represents the true value, $h(z_i)$ is the predicted value, and $\mathcal{L}(h)$ denotes the loss function.

### 3.7 Hard-voting or majority-voting

Hard-voting ensemble methods combine predictions from multiple ML models to produce a final prediction [48]. This process can be expressed using the following equation:

$$\sum_{j=1}^{N} D_{j,k} = \max_k \sum_{j=1}^{N} D_{j,k} \tag{14}$$

Here, $k = 1, 2, 3, \ldots$ represents the classes, and $j = 1, 2, 3, \ldots, N$ indicates the different ML models. The term $D_{j,k}$ refers to the decision output from the $j$-th model for class $k$. The expression $\max_k \sum_{j=1}^{N} D_{j,k}$ identifies the class $k$ that receives the highest total number of votes from all models.

### 3.8 Soft-voting or weighted average

Soft-voting is a method where the predicted probabilities of each class from individual ML models are averaged to decide the final class label [49]. Mathematically, this can be described by the following equation:

$$\hat{y} = \frac{\sum_{j=1}^{N} v_j g_j(z)}{\sum_{j=1}^{N} v_j} \tag{15}$$

This equation, $\hat{y}$ denotes the predicted output or label. $N$ represents the total number of independent ML models or base learners in the ensemble. The variable $v_j$ indicates the weight assigned to the prediction of the $j^{th}$ model. In this framework, the prediction from each base learner is scaled by its respective weight. The term $\sum_{j=1}^{N} v_j g_j(z)$ aggregates the weighted predictions from each model $g_j(z)$ for a given input $z$. The denominator, $\sum_{j=1}^{N} v_j$, represents the total sum of the weights, which is used to normalize the weighted sum of predictions.

### 3.9 Stacking

Stacking, also known as stacked generalization, is an ensemble technique that combines multiple ML models, or base learners, to improve predictive accuracy [50]. This method involves training a meta-learner or a higher-level model, which learns the best way to aggregate the predictions from different base models. Meta-learning is a specialized form of learning where algorithms are trained using the outputs of other ML algorithms to produce more accurate predictions.

### 3.10 Blending

Blending is an ensemble learning approach where a separate machine learning model is trained to optimally combine the predictions of multiple base models [51]. In ML, blending is a meta-ensemble method that combines predictions from multiple base models using a secondary model to produce the final predictions.

## 4 Proposed approach

This paper proposes a hybrid ML model designed for efficient electrical fault analysis and classification. The framework of the proposed hybrid model is illustrated in Fig 1. It encompasses several key phases: data collection, data preprocessing, data splitting, hybrid model integration, and classification. The process begins with collecting and preprocessing data, which includes label encoding and standard scaling. The dataset is then split into 85% for training and 15% for testing. The hybrid model-1 *(RF + DT + Stacking)* with GB as the meta-learner to enhance predictive accuracy. This model leverages the strengths of RF and DT, combined through GB, to refine predictions and improve performance. The final classification phase assesses the model's effectiveness using evaluation metrics such as accuracy, precision, recall, and F1 score. Algorithm 1 outlines the steps of the proposed hybrid approach. The model combines Random Forest (RF) and Decision Tree (DT) as base learners, whose predictions are then stacked and fed into a Gradient Boosting (GB) meta-learner to produce the final classification. This ensemble strategy leverages the complementary strengths of RF and DT while using GB to refine and optimize the final prediction.

### 4.1 Dataset

The dataset used for this research is the Electrical Faults Analysis & Classification dataset, which comprises a total of 7861 samples with ten distinct features [52]. This dataset was generated through simulation experiments in MATLAB/Simulink, modeling electrical transmission lines under diverse operating and fault conditions. The data was recorded at a sampling rate of 10 kHz, ensuring accurate capture of transient fault events. Fig 2 illustrates the relative importance of each feature used in the classification task. The ordinate axis ("Importance") represents the normalized feature importance scores, computed using the Gini impurity criterion from the Random Forest (RF) algorithm. These scores quantify

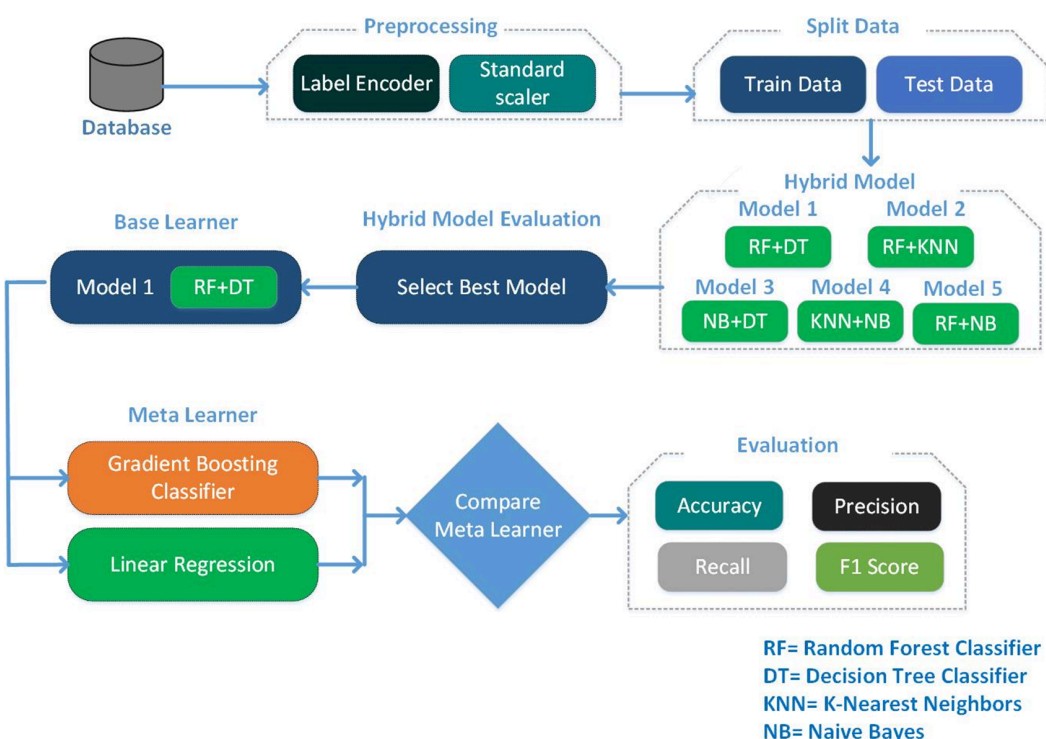

**Fig 1**. **Proposed methodology.**

**Algorithm 1.   Hybrid Stacking Ensemble Model for Electrical Fault Detection (Base Learners: Random Forest & Decision Tree; Meta-Learner: Gradient Boosting).**

**Input:** Input dataset $\mathbf{X}$ with feature vectors $\mathbf{x}_i$ and labels $\mathbf{y}_i$
**Output:** Trained hybrid model capable of predicting fault classes

1 **Initialize parameters and hyperparameters:** $N_{RF}$ (Number of trees in Random Forest), DT_depth (Max depth of Decision Trees), and GB parameters (e.g., learning rate, number of boosting stages);

2 **Random Forest Training:** Initialize $N_{RF}$ decision trees;

3 **for** *each decision tree i in Random Forest* **do**

4    Train DT$_i$ using $\mathbf{X}$ and $\mathbf{y}$;

5 **end**

6 **Decision Tree Training:** Train a single Decision Tree DT using $\mathbf{X}$ and $\mathbf{y}$, with depth DT_depth;

7 **Stacking Ensemble Training with GB:** Generate predictions from Random Forest RF($\mathbf{X}$) and Decision Tree DT($\mathbf{X}$);

8 **Initialize GB model:**  Train GB model on concatenated predictions:

$$\hat{\mathbf{y}} = \mathrm{GB}([\mathrm{RF}(\mathbf{X}), \mathrm{DT}(\mathbf{X})])$$

9 **Prediction Phase: for** *each new input vector* $\mathbf{x}_{new}$ **do**

10    Obtain predictions from Random Forest and Decision Tree:

$$\mathrm{RF}(\mathbf{x}_{new}), \quad \mathrm{DT}(\mathbf{x}_{new})$$

   Feed predictions into trained GB model to obtain final prediction:

$$\hat{y}_{new} = \mathrm{GB}([\mathrm{RF}(\mathbf{x}_{new}), \mathrm{DT}(\mathbf{x}_{new})])$$

11 **end**

12 **Performance Evaluation:** Evaluate the model's performance using metrics like accuracy, precision, recall, and F1-score;

13 **Parameter Tuning:** Optimize model hyperparameters using techniques like grid search or Bayesian optimization;

14 **Return:** the optimized hybrid ML model capable of predicting transmission line fault classes;

the contribution of each feature to reducing classification error across all decision trees in the ensemble. The results indicate that current-based features (Ia, Ib, Ic) exhibit higher importance compared to voltage-based features (Va, Vb, Vc). This is expected because fault events typically cause sharper deviations in current signals, making them more discriminative for classification, whereas voltage signals tend to vary more gradually and thus provide relatively lower predictive power. Fig 3 displays the proportion of samples corresponding to each fault category within the dataset. Before model training, data preprocessing was performed to ensure consistency and reliability. This included consolidating the output labels into a single column, removing missing or inconsistent values, and normalizing all features to a uniform scale. Since the dataset already contained diagnostic attributes directly related to fault classification, no additional handcrafted feature engineering was required. For clarity, we combined the output features into a single column, representing the fault type. Table 2 summarizes the fault types and their corresponding output representations. It should be noted that the "No-Fault" regime is included as a reference category to enable the model to distinguish between healthy and faulty system states.

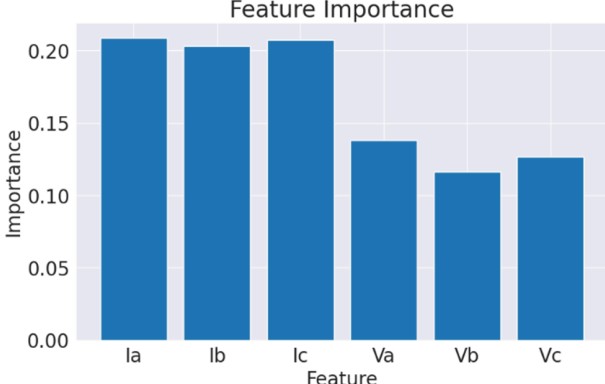

**Fig 2**. **Importance of features for classifying the fault.**

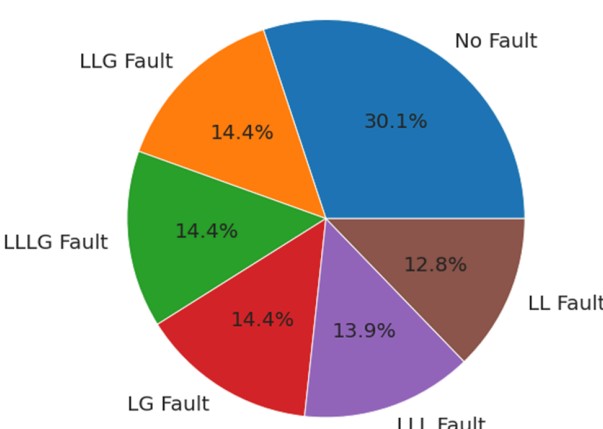

**Fig 3**. **Percentage distribution of fault types in the dataset.**

**Table 2**. **The experimental results of the Hybrid models (Base ML Models + Ensemble Techniques) for fault classification.**

| Output Representation | Fault Type |
|---|---|
| [0 0 0 0] | No-Fault |
| [1 0 0 1] | LG Fault (Between Phase A and Ground) |
| [0 0 1 1] | LL Fault (Between Phase A and Phase B) |
| [1 0 1 1] | LLG Fault (Between Phases A, B, and Ground) |
| [0 1 1 1] | LLL Fault (Short-circuit between all three phases without ground involvement.) |
| [1 1 1 1] | LLLG Fault (Symmetrical fault involving all three phases and ground.) |

The dataset covers six classes (No-Fault, LG, LL, LLG, LLL, LLLG), representing the most frequent and severe transmission line faults. However, other fault scenarios such as Phase B–Ground, Phase C–Ground, or inter-phase faults (A–C, B–C) are not explicitly included in this dataset. This limitation arises from the original dataset design and is acknowledged as a potential extension for future work to enhance the generalizability of the proposed model.

This classification framework allows for accurate and efficient identification of six distinct types of faults, essential for effective fault detection and maintenance planning in electrical power systems. The simulated nature and diversity of the dataset provide a comprehensive basis for training and evaluating the proposed hybrid model.

## 4.2 Data preprocessing

Data preprocessing is a fundamental step in preparing the dataset for ML models. In our dataset, some features have categorical data, and some feature numeric data and overfitting problems. So, in this research, we have used Label Encoding and Standard Scaler to suit the data [53], ultimately enhancing the performance and reliability of the model. Label encoding is a technique for converting categorical data into numerical format. It assigns a unique integer to each category, allowing ML algorithms to process the data. On the other hand, Standard Scaler is also a data preprocessing technique that transforms numerical features into a mean of 0 and a standard deviation of 1. This standardization process is crucial for many ML algorithms, ensuring that features with different scales contribute equally to the model's learning process.

## 4.3 Hybrid model

This study developed hybrid models by combining various supervised ML algorithms to leverage their strengths and improve overall predictive performance. Each hybrid model integrates two different algorithms, providing a diverse approach to classification. Five unique hybrid models were created by pairing various ML algorithms to identify the most effective combinations. These combinations are detailed in Table 3.

Subsequently, these hybrid models were integrated with four ensemble techniques: Soft-Voting, Hard-Voting, Stacking, and Blending. Using these combinations, we constructed various ensemble ML models. Comprehensive experiments were conducted on these ensemble models (hybrid models + ensemble techniques) to identify the optimal ensemble model for fault classification tasks. Based on the experimental results, we identified the most influential ensemble model for electrical fault analysis as the combination of RF, DT, and Stacking, denoted as *RF+DT+Stacking* and achieved an Accuracy of 93.64%.

In Model-1, the RF and DT algorithms were combined to balance complexity and interpretability. RF, an ensemble method, excels at capturing intricate patterns and reducing overfitting by aggregating multiple decision trees, while DT offers transparency with its hierarchical decision-making. Similarly, the other models also integrate complementary algorithms in order to enhance classification performance. Model-2 (RF + KNN) combines RF's robustness with KNN's instance-based learning. RF constructs multiple DT to extract deep feature representations, while KNN classifies instances based on their similarity to neighboring samples. This combination enhances adaptability in complex classification tasks by leveraging RF's ability to handle high-dimensional data and KNN's effectiveness in local pattern recognition. whereas Model-3 (NB + DT) leverages NB's probabilistic approach with DT's interpretability. NB efficiently handles categorical data and assumes feature independence to make it computationally efficient, while DT enables structured decision-making. The fusion of these classifiers enhances predictive accuracy by incorporating both probabilistic reasoning and rule-based learning. In contrast, Model-4 (KNN + NB) merges KNN's local generalization with NB's simplicity. KNN determines class membership by evaluating the proximity of neighboring samples, while NB estimates class

**Table 3**. **List of Hybrid models using various combinations of ML algorithms.**

| Hybrid Model | ML Algorithms |
| --- | --- |
| Model-1 | RF + DT |
| Model-2 | RF + KNN |
| Model-3 | NB + DT |
| Model-4 | KNN + NB |
| Model-5 | RF + NB |

probabilities based on feature distributions. The combination enhances classification robustness, particularly in scenarios with varying data distributions and noise. Finally, Model-5 (RF + NB) fuses RF's ensemble strength with NB's probabilistic nature. RF enhances feature extraction through its diverse DT's, while NB provides a lightweight and efficient classification mechanism. This integration aims to improve generalization by combining RF's feature importance ranking with NB's fast probabilistic inference. As a result, these combinations ensure diverse and improved fault classification outcomes.

## 4.4 Meta learner

The meta-learner serves as an integrative layer, trained on the predictions of base learners rather than directly on the original data. Acting as a higher-level model, the meta-learner learns the relationships between base model predictions and the actual target values, refining the final predictions by effectively combining base learners' outputs. Common choices for meta-learners include linear regression (LR), logistic regression, and gradient-boosting (GB). By leveraging the strengths of diverse base models and mitigating their weaknesses, the meta-learner aims to improve the final predictive model's accuracy, robustness, and generalization. This study used LR and GB as meta-learners to aggregate predictions from various base models, and we selected GB as the preferred meta-learner due to its superior performance in Model-1, where it achieved the highest accuracy. By leveraging its iterative approach to refine predictions and capture complex patterns, GB enhances the stacking ensemble's accuracy and robustness, significantly improving linear regression.

## 4.5 Evaluation metrics

To evaluate the proposed hybrid ML model's performance, critical metrics such as True Positive (TP), False Positive (FP), True Negative (TN), and False Negative (FN) are derived from the confusion matrix. These metrics are then utilized to compute Precision, Recall, F1-score, and Accuracy. This comprehensive assessment framework allows for a thorough evaluation of the model's overall effectiveness in classification tasks [54,55].

Let $M$ be the number of classes and $q_i$ be the support (number of true samples) for class $i$. Define true positives $TP_i$, false positives $FP_i$, and false negatives $FN_i$ for class $i$. Total samples: $|Q| = \sum_{i=1}^{M} q_i$.

**Accuracy**

$$A = \frac{1}{|Q|} \sum_{i=1}^{M} TP_i \tag{16}$$

**Per-class Precision, Recall, F1**

$$P_i = \frac{TP_i}{TP_i + FP_i}, \quad R_i = \frac{TP_i}{TP_i + FN_i}, \quad F1_i = \frac{2P_i R_i}{P_i + R_i} \tag{17}$$

**Weighted averages**

$$P_w = \frac{1}{|Q|} \sum_{i=1}^{M} P_i q_i, \quad R_w = \frac{1}{|Q|} \sum_{i=1}^{M} R_i q_i, \quad F1_w = \frac{1}{|Q|} \sum_{i=1}^{M} F1_i q_i \tag{18}$$

## 5 Experiment results

In this section, we present the results of our experiments designed to evaluate the proposed hybrid model's performance. We assess the model's effectiveness in classifying transmission line faults using various performance metrics, including accuracy, precision, recall, and F1-score.

## 5.1 Result

In this research, we conducted a comprehensive experiment. Initially, we assess the performance of various ML algorithms for Electrical fault classification tasks using the Electrical Faults Analysis & Classification dataset. Subsequently, we conduct experiments involving five hybrid ML models in Table 3 and five ensemble techniques on the datasets to determine the optimal ensemble model for electric fault classification. Table 4 presents the performance metrics of various ML algorithms evaluated in our study. The table includes A, P, R, and F1-Score for each algorithm, comprehensively comparing their effectiveness in classifying electrical faults. The DT and RF algorithms both achieved the highest $A$ of 88.73%, demonstrating their robustness in fault classification tasks. We note that the accuracies of the DT and RF algorithms are equal. Therefore, we consider memory usage and computational complexity as additional factors for evaluation. The execution time and memory of DT is less than RF. However, the DT slightly outperformed the RF in terms of time and memory. The KNN algorithm achieved an $A$ of 78.31%, with $P$ and $F1–Score$ slightly lower, reflecting its moderate performance in this context. The SVM algorithm had the lowest $A$ of 74.32% and $F1–Score$ of 72.25%, indicating limited effectiveness for this specific dataset. NB achieved a better performance than KNN and SVM with an $A$ of 79.75%, and a balanced $F1–Score$ of 76.66%. AdaBoost performed slightly better than NB with an $A$ of 78.81% and a $P$ of 77.29%. These results highlight that tree-based algorithms (DT, RF) are inherently more effective for this type of fault data because they capture non-linear relationships and handle categorical features efficiently, whereas distance-based (KNN) and margin-based (SVM) classifiers struggle due to overlapping fault patterns in the dataset. From a real-time applicability perspective, NB and DT exhibit the lowest execution times (62 ms and 205 ms, respectively) and minimal memory requirements, making them highly suitable for online grid fault detection systems. RF, despite its competitive accuracy, requires significantly higher execution time (2524 ms) and memory (13.67 MB), which may limit its deployment in latency-sensitive environments. Similarly, algorithms like SVM and AdaBoost demonstrate higher computational overheads, raising concerns for real-time grid monitoring. Therefore, while accuracy remains an important criterion, execution time and memory efficiency suggest that DT and NB can serve as practical candidates for real-time deployment, whereas RF and SVM may be more appropriate in offline or batch-processing scenarios.

Table 5 presents the classification results of the hybrid models, which combine base ML models with Hard Voting, Soft Voting, Stacking, and Blending techniques for fault classification tasks. Model-1 exhibited the best overall performance, achieving the highest $A$, $P$, $R$, and $F1–Score$ values of 93.64%, 93.65%, 93.64%, and 93.64%, respectively, with the Stacking (Gradient Boosting) technique. This result highlights its robustness in fault classification tasks. Compared with the single models, the hybrid models consistently improved accuracy by 4–6%, which confirms that ensemble learning reduces the variance and bias of individual classifiers. Among them, Stacking with Gradient Boosting provided the best balance between predictive accuracy and stability. Similarly, Model-1 outperformed other models with the Hard Voting technique, obtaining $A$, $P$, $R$, and $F1–Score$ values of 88.56%, 88.58%, 88.56%, and 88.55%. In the Soft Voting technique, Model-1 also achieved the best results, with $A$, $P$, $R$, and $F1–Score$ values of 89.49%, 89.50%, 89.49%,

**Table 4**. Performance metrics of various ML algorithms.

| ML Algorithm | A | P | R | F1 | Time (ms) | Memory (MB) |
|---|---|---|---|---|---|---|
| DT | 0.89 | 0.89 | 0.89 | 0.89 | 205 | 1.04 |
| RF | 0.89 | 0.89 | 0.89 | 0.89 | 2524 | 13.67 |
| KNN | 0.78 | 0.78 | 0.78 | 0.78 | 136 | 0.63 |
| SVM | 0.74 | 0.71 | 0.74 | 0.72 | 1468 | 8.33 |
| NB | 0.80 | 0.79 | 0.80 | 0.77 | 62 | - |
| AdaBoost | 0.79 | 0.77 | 0.79 | 0.76 | 957 | 0.03 |

**Table 5.** The experimental results of the Hybrid models (Base ML Models + Ensemble Techniques) for fault classification.

| Hybrid Models | Hard Voting | | | Soft Voting | | | Stacking (Linear Regression) | | | Stacking (Gradient Boosting) | | | Blending | | |
|---|---|---|---|---|---|---|---|---|---|---|---|---|---|---|---|
| | P | A | F1 | P | A | F1 | P | A | F1 | P | A | F1 | P | A | F1 |
| **Model-1** | **0.89** | **0.89** | **0.89** | **0.90** | **0.90** | **0.90** | **0.92** | **0.91** | **0.91** | **0.94** | **0.94** | **0.94** | **0.73** | **0.72** | **0.66** |
| Model-2 | 0.81 | 0.82 | 0.81 | 0.84 | 0.84 | 0.84 | 0.85 | 0.85 | 0.85 | 0.93 | 0.93 | 0.93 | 0.70 | 0.73 | 0.68 |
| Model-3 | 0.80 | 0.81 | 0.77 | 0.89 | 0.89 | 0.89 | 0.89 | 0.89 | 0.89 | 0.87 | 0.87 | 0.87 | 0.64 | 0.73 | 0.67 |
| Model-4 | 0.75 | 0.80 | 0.76 | 0.77 | 0.78 | 0.77 | 0.90 | 0.90 | 0.90 | 0.91 | 0.91 | 0.91 | 0.65 | 0.74 | 0.68 |
| Model-5 | 0.77 | 0.80 | 0.76 | 0.85 | 0.85 | 0.85 | 0.84 | 0.84 | 0.84 | 0.93 | 0.93 | 0.93 | 0.87 | 0.88 | 0.88 |

and 89.48%, respectively. Moreover, Model-1 outperformed other models with the Stacking (Linear Regression) technique, achieving $A$, $P$, $R$, and $F1$–Score values of 91.44%, 91.46%, 91.44%, and 91.43%, respectively. Similarly, Model-5 excelled in the Blending technique, achieving $A$, $P$, $R$, and $F1$–Score values of 87.54%, 87.39%, 87.54%, and 87.46%, respectively.

Other hybrid models also demonstrated notable results, with Model-2 and Model-3 showing competitive outcomes under the Soft Voting technique, while Model-4 and Model-5 achieved moderate performances across both techniques.

Fig 4 illustrates the accuracy of the hybrid ML models. Model-1 achieved the highest accuracy, reaching 93.64%, particularly with the Stacking, reinforcing the earlier observation. This consistent outperformance across ensemble methods indicates that carefully combining decision-tree-based learners can yield more reliable classification than relying on individual algorithms.

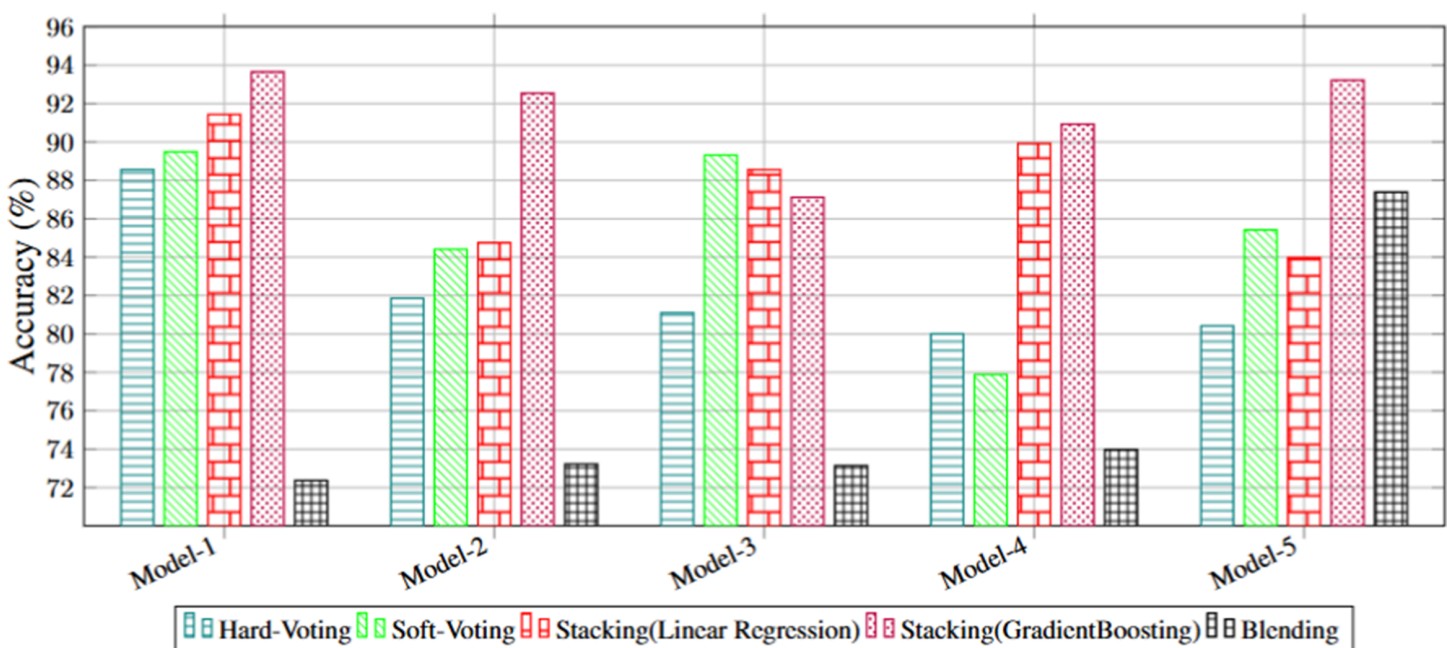

**Fig 4.** Electrical fault classification accuracy of the Hybrid ML models.

## 6 Discussion

In the field of electrical fault classification, researchers have proposed numerous ML and hybrid models to enhance classification accuracy, highlighting ample room for improvement in model performance. We address the research problem "***Which hybrid machine learning model can outperform previous research and improve the accuracy of electrical fault classification?***" by developing various hybrid ML models and conducting extensive experiments. We utilized six ML algorithms—namely, KNN, DT, RF, AdaBoost, NB, and SVM, both independently and in various combinations to construct hybrid ML models. Additionally, we employed four ensemble techniques, including stacking, blending, hard-voting, and soft-voting. Prior to experimentation with individual ML algorithms, we executed data preprocessing tasks and observed a substantial impact on classification performance. Table 4 displays the classification performance of the ML algorithms, with DT and RF achieving the highest $A$ of 89% and 89%, respectively. Table 5 showcase the performance of ensemble models. The ensemble ML model (Model-1 + Stacking) achieved the highest $A$, $P$ and $F1–Score$ of 94%, and Model-1 exhibited commendable performance with other ensemble techniques. Although execution time and memory consumption were analyzed to assess computational efficiency, the proposed framework has not been validated on IEEE benchmark systems or real-world operational data. Therefore, claims related to real-time deployment should be interpreted as prospective rather than experimentally demonstrated. Nevertheless, the lightweight nature of the proposed ensemble suggests its suitability for future real-time implementation, subject to further validation on standardized test systems and field data.

Table 6 illustrates the performance comparison between previous research and our proposed hybrid model for the electrical fault classification task. Our proposed work achieves approximately 94% accuracy, which is slightly lower than the other methods presented in the literature. The work [56] achieves 99% accuracy with a Kalman filter and a second-order low-pass filter, specifically designed for Active Distribution Networks (ADNs) with bidirectional power flows. The study [57], with 90% accuracy, focuses on fault isolation in smart grids using complex current and voltage criteria, tested on the IEEE 13-node test feeder. In [58] achieves an impressive 99.93% accuracy using a hybrid deep learning model that combines wavelet packet transform and LSTM, specifically designed for photovoltaic systems. The paper [59], with 98.85% accuracy, utilizes a CNN optimized by Gorilla Troops Optimization (GTO) for microgrids, effectively detecting, classifying, and locating faults. While deep learning methods outperform our model in raw accuracy, they typically require significantly larger datasets, high-end computational resources, and domain-specific feature engineering. In contrast, our proposed hybrid model achieves competitive performance on a modest dataset while remaining lightweight, interpretable, and computationally efficient. Although ML techniques can theoretically be applied to any tabular dataset, electrical fault classification is unique because the features arise directly from power-system transient dynamics. The model must learn system-specific nonlinearities resulting from impedance, line length, and fault resistance, which are not present in generic classification tasks. This makes the proposed approach particularly suitable for resource-constrained environments and rapid deployment in practical power system monitoring. When we used the binary class dataset, our model achieved an accuracy of 99.58% [52]. The superior results suggest that the proposed hybrid model is effective for electrical fault classification tasks.

**Table 6**. **Performance metrics of various ML algorithms.**

| SL. | Reference | Year | Model | Fault Class | Accuracy |
|---|---|---|---|---|---|
| 01 | [56] | 2020 | Hybrid Model | 4 | 99% |
| 02 | [57] | 2022 | dual-stage DKF and SOLPF | 4 | 90% |
| 03 | [58] | 2022 | Hybrid Deep Learning | - | 99.93% |
| 04 | [59] | 2022 | CNN-GTO | 4 | 98.85% |
| 05 | Proposed | 2025 | Hybrid Model (RF + DT + Stacking) | 6 | 93.64% |
| 06 | Proposed | 2025 | Hybrid Model (RF + DT + Stacking) | 2 | 99.58% |

## 7 Conclusion

Fault classification in electrical systems has become a crucial area of research, particularly with the growing complexity of modern power grids and the need for reliable system protection. In this study, we developed five hybrid models using advanced ensemble techniques such as Hard-Voting, Soft-Voting, Stacking, and Blending to determine the most effective approach for fault classification. Extensive experiments were conducted using electrical fault datasets to evaluate the performance of these models. Based on the experimental findings, the hybrid model denoted as (RF+DT+Stacking) achieved the highest classification accuracy of 93.64%, outperforming other ensemble models. Compared with prior works, our model offers competitive accuracy while maintaining computational efficiency. For example, deep learning-based methods in the literature achieve very high accuracy (98–99.9%), but they typically require extensive datasets and higher computational resources. By contrast, our model demonstrates a more practical trade-off between performance and efficiency, making it more feasible for real-time grid monitoring. However, this study also has several limitations. The dataset used is relatively small and primarily simulated, which may not fully capture the diversity of real-world grid conditions. Additionally, the evaluation does not include large-scale IEEE benchmark systems or extensive statistical robustness tests (e.g., cross-validation). For future work, we plan to: (i) validate the model on larger and more diverse real-world datasets; (ii) apply the method to IEEE benchmark test systems; (iii) incorporate explainable AI methods to improve interpretability for operators; and (iv) explore additional hybrid and deep learning architectures to further enhance classification accuracy and robustness.

## 8 Strengths, limitations, and future perspectives

This research offers a robust hybrid ML model combining RF, DT, and Stacking techniques, significantly improving fault classification accuracy. The inclusion of ensemble techniques enhances decision-making, contributing to more reliable and proactive grid maintenance. The model's high-performance metrics demonstrate its effectiveness in handling complex transmission line faults.

This study evaluates model performance using a single train–test split (85%–15%) due to computational and time constraints. Consequently, cross-validation (e.g., k-fold validation), robustness analysis under varying noise levels, and statistical significance testing were not performed. While the reported results provide useful comparative insights, they should not be interpreted as definitive evidence of strong generalization across unseen operating conditions. Future work will incorporate cross-validation, sensitivity analysis, and statistical tests to strengthen the reliability of the conclusions.

The model's performance heavily depends on the dataset's quality and diversity. Additionally, the complexity of the hybrid approach increases computational requirements, potentially limiting real-time applications in resource-constrained environments.

Future research could focus on improving computational efficiency and exploring deep learning methods for more nuanced fault classification. Expanding the dataset to cover various environmental and operational conditions will enhance the model's adaptability and robustness in diverse grid scenarios.

## Author contributions

**Conceptualization:** Momotaz Begum, Ariful Islam Shiplu, Mehedi Hasan Shuvo, Jia Uddin.

**Data curation:** Momotaz Begum.

**Formal analysis:** Ariful Islam Shiplu, Fahmid Al Farid.

**Methodology:** Momotaz Begum, Ariful Islam Shiplu, Mehedi Hasan Shuvo, Sumaiya Ismat Jerin.

**Resources:** Mehedi Hasan Shuvo, Fahmid Al Farid.

**Supervision:** Hezerul bin Abdul Karim.

**Validation:** Momotaz Begum, Jia Uddin.

**Visualization:** Ariful Islam Shiplu, Mehedi Hasan Shuvo, Jia Uddin.

**Writing – original draft:** Momotaz Begum, Ariful Islam Shiplu, Mehedi Hasan Shuvo, Fahmid Al Farid, Sumaiya Ismat Jerin, Jia Uddin.

**Writing – review & editing:** Hezerul bin Abdul Karim.

**Funding:** Hezerul bin Abdul Karim.

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
