## [Decision Letter · Decision Letter 0]

1 Aug 2025

PONE-D-25-30171Enhancing Reliability in Electrical Grids: A Hybrid Machine Learning Approach for Electrical Faults ClassificationPLOS ONE

Dear Dr. Uddin,

Thank you for submitting your manuscript to PLOS ONE. After careful consideration, we feel that it has merit but does not fully meet PLOS ONE’s publication criteria as it currently stands. Therefore, we invite you to submit a revised version of the manuscript that addresses the points raised during the review process.

We look forward to receiving your revised manuscript.

Kind regards,

Haris Calgan, Ph.D.

Academic Editor

PLOS ONE

Journal Requirements:

“This research was funded by Multimedia University, Cyberjaya, Selangor, Malaysia

(Grant Number: PostDoc (MMUI/240029)).”

3. Please note that your Data Availability Statement is currently missing the repository name. If your manuscript is accepted for publication, you will be asked to provide these details on a very short timeline. We therefore suggest that you provide this information now, though we will not hold up the peer review process if you are unable.

5. We note you have included a table to which you do not refer in the text of your manuscript. Please ensure that you refer to Table 1 in your text; if accepted, production will need this reference to link the reader to the Table.

Additional Editor Comments:

Enclosed please find detailed reviewer reports on the paper that you have submitted for possible publication in the PLOS ONE.

The reviewers require a MAJOR REVISION of your paper.

If you are prepared to undertake the work required, I would be pleased to reconsider my decision. However, I would like to ask you to very carefully address the issues raised by the reviewers and revise your paper accordingly.

Please keep in mind that if citation suggestions have been made by reviewers, these suggestions are optional and will not influence the editor's decision. Do not include citations that are irrelevant to the paper or that you believe will not add value to the manuscript.

Reviewers' comments:

Reviewer's Responses to Questions

**Comments to the Author**

1. Is the manuscript technically sound, and do the data support the conclusions?

Reviewer #1: Partly

Reviewer #2: No

Reviewer #3: Partly

Reviewer #4: Yes

Reviewer #5: Partly

2. Has the statistical analysis been performed appropriately and rigorously?

Reviewer #1: No

Reviewer #2: No

Reviewer #3: Yes

Reviewer #4: N/A

Reviewer #5: I Don't Know

3. Have the authors made all data underlying the findings in their manuscript fully available?

Reviewer #1: No

Reviewer #2: Yes

Reviewer #3: Yes

Reviewer #4: Yes

Reviewer #5: Yes

4. Is the manuscript presented in an intelligible fashion and written in standard English?

Reviewer #1: No

Reviewer #2: No

Reviewer #3: Yes

Reviewer #4: Yes

Reviewer #5: No

5. Review Comments to the Author

Reviewer #1: 1. The paper lacks in major contribution and motivation of research.

2. The background and significance of this study should be highlighted in the abstract.

3. Check the English presentation of this paper to remove the typo mistakes. Some grammatical issues need to be addressed in the whole text. Please reform the long paragraphs. Please polish the writing and English of the manuscript carefully. The writing of the paper needs a lot of improvement in terms of grammar, spelling, and presentation. The paper needs careful English polishing since there are many typos and poorly written sentences. I found several errors.

4. In the "Introduction" section, a more detailed analysis of the existing literature on the subject is needed, and an in-depth analysis of the possible application fields.

5. The mathematics used throughout the article is still not very strict. Please try to update and illustrate some elements in the mathematical model that are not defined very strictly.

6. The overall structure of the article should be improved.

7. The result part is week, results and discussion should be better explained.

8. References must be updated and add the suitable from the following

https://doi.org/10.3390/en12050961, https://doi.org/10.1109/EIConRus.2018.8317170, https://doi.org/10.1016/j.est.2024.113556, https://doi.org/10.1109/ACCESS.2024.3437191, https://doi.org/10.1109/MEPCON.2017.8301313, https://dx.doi.org/10.21608/jaet.2021.82231, https://doi.org/10.1109/MEPCON47431.2019.9008171, https://doi.org/10.21608/sej.2021.155557,

https://doi.org/10.1109/MEPCON58725.2023.10462371, DOI: 10.1109/ACCESS.2024.3525183, https://doi.org/10.1007/s00521-024-09433-3, https://doi.org/10.1371/journal.pone.0317619, https://doi.org/10.20508/ijrer.v14i2.14346.g8898, https://doi.org/10.20508/ijrer.v13i1.13718.g8659, https://doi.org/10.1007/s00521-024-09433-3, https://doi.org/10.1007/s00521-024-09902-9, https://doi.org/10.21608/SVUSRC.2024.279389.1198,

9. Check all of your Figures and Tables have a good explanation of your text.

10. Many paragraphs without citations

11. What are the contributions and novelty of work mentioned?

12. The authors' conclusions need to be improved, a comparison of the results obtained with those already existing in the literature would be appropriate. I suggest also describing what can still be improved in this work, which can still be improved based on the results obtained, according to the authors' view. It is suggested to offer some limitations existed in this study and an outlook for future study in the last section.

Reviewer #2: The manuscript presents a hybrid machine learning model for fault classification in electrical transmission lines using a combination of RF, DT, and stacking techniques. However, the proposed approach lacks novelty, relies heavily on standard methods, and does not provide sufficient technical rigor or validation to justify publication. The paper also fails to engage with recent literature and presents overstated contributions without empirical support.

Reviewer Comments:

1.The hybrid ensemble model proposed in this work (RF + DT + Gradient Boosting) is a routine ML pipeline with no algorithmic innovation or novel architecture. Similar stacking-based models have already been widely reported in existing literature.

2.The dataset used is publicly available and small in scale, with no cross-validation, statistical tests, or robustness analysis. This undermines the reliability and generalizability of the reported 93.64% accuracy.

3.References are outdated and do not include any recent (2023–2025) high-impact studies using advanced techniques such as deep learning, GNNs, or explainable AI in power systems. This severely limits the relevance of the work.

4.Many sections are filled with textbook-level explanations of basic ML algorithms (e.g., Naive Bayes, KNN), which do not contribute to the research and make the manuscript overly lengthy without adding scientific value.

5.The manuscript lacks real-world validation on standard IEEE test systems, and key aspects such as model interpretability, class-wise performance, and computational feasibility in grid environments are not addressed.

Reviewer #3: This is a solid and well-written paper that adds to the body of knowledge about how to use machine learning to identify faults in power systems. The design of the experiment is strong, the results are shown, and the open data/code policy makes it easier to reproduce the results.

The paper says that the hybrid model is new; however, other research has used similar stacking ensembles to find faults (see [15], [16], and [22]). The exact combo of RF+DT+GB works, but it isn't really new.

There is a difference in the stated accuracy. The abstract and Section 4.2 say 93.64%, whereas Table 5 shows 94% for Model-1 with Stacking (GB), and Table 6 includes 93.64% and 99.58% (binary example).

Section 5 of the discussion says, "Our proposed work achieves about 94% accuracy, which is a little lower than the other methods shown in the literature." Table 6, on the other hand, demonstrates that some earlier works had better accuracy (98–99.93%).

The report says that GB was chosen over linear regression because it worked better; however, it doesn't show LR's results for comparison.

The dataset comes from Kaggle [35], but there are no details on how the data was created (simulation, real-world, lab setting), the sampling rate, or feature engineering.

Table 4 shows execution time and memory, but the text doesn't talk about how it can be used in real time.

A cognitive science definition of "blending" is given first, which is not useful in the context of ML.

Figure 4 Please put "Accuracy" on the y-axis and make sure that all the numbers match those in Table 5.

Equations, Make sure the formatting is the same throughout (for example, Equation 14 uses f(x) and f^(x), but the text uses y^).

Table 6, To make the contrast between 6 classes and 2 classes clearer, add a column for "Fault Classes."

In the abstract, don't list all of the machine learning algorithms; just say "various machine learning algorithms."

Conclusion: Don't rehash the results; instead, the conclusion should bring together the implications.

Reviewer #4: The answers to the questions above and the issues requiring further clarification and improvement in the paper:

1.Technical Soundness & Conclusion Support (Q1: Yes)

The manuscript presents a technically sound study, and the experimental/data analysis methods are appropriate for addressing the research question. The conclusions are supported by the results. However, In order to further strengthen the conclusion that the proposed hybrid model is effective for electrical fault classification tasks, I suggest to conduct a comparative analysis of the four evaluation indexes in the discussion section.

2.Statistical Analysis (Q2: N/A)

The question regarding statistical analysis is marked as 'N/A' because this study focuses on evaluating the performance of a hybrid machine learning model for fault diagnosis, where the results are primarily validated through computational metrics (e.g., accuracy, precision, F1-score) rather than traditional statistical hypothesis testing. If the authors employed any statistical methods (e.g., significance testing of model comparisons), these should be explicitly stated in the Methods section.

3.Data Availability (Q3: Yes)

The manuscript utilizes publicly available data from the Electrical Faults Analysis & Classification dataset, and the source is appropriately cited in Section 3.1. If the database requires specific licensing statements, these should be added.

4.Language & Clarity (Q4: Yes)

This paper is written in standard English and has a relatively clear logical structure. However, some parts of Algorithm 1 and 4.1 need to be appropriately explained and adjusted according to the overall research ideas of the paper.

5.Model combination strategy lacks comprehensiveness

The authors selected DT and RF for combination based solely on their individual performance. While this approach is intuitive, it lacks a systematic evaluation of all possible model pairs. Other combinations (such as SVM and NB) may provide complementary strengths that lead to better hybrid performance. I suggest including comparative experiments on all pairwise model combinations to ensure the selected hybrid model is indeed optimal.

6.Model selection rationale needs further justification

Although the study focuses on A,P,R,F1, execution time and memory usage are also reported but not analyzed. If the hybrid of DT and RF is not the best in terms of time and memory consumption, the rationale behind choosing this pair should be elaborated. For example, is model interpretability, robustness, or ease of deployment a contributing factor? Currently, the reasoning appears somewhat incomplete.

7.Fairness of Comparative Experiments

It is recommended that the authors clarify whether all models were evaluated under

consistent experimental conditions—such as using the same training/testing splits and comparable hyperparameter optimization strategies. Ensuring uniform settings is essential for a fair and valid comparison of model performance.

8.Generalizability of the Hybrid Approach

The current hybrid modeling strategy may be tailored to the specific characteristics of the

dataset or fault type used in this study. The authors are encouraged to discuss whether the proposed method is generalizable to other fault diagnosis scenarios or industrial datasets, and to specify any potential limitations regarding its applicability.

9. manuscript structure

The evaluation metrics (e.g.,accuracy, precision, recall, F1-score Equations (15)-(17)) are currently introduced and defined in Section 4.1. However, these metrics are integral to the methodology and should ideally be presented earlier in the paper—preferably in the section describing the diagnostic framework or evaluation methodology. This would improve the logical flow of the manuscript.

Reviewer #5: One of the consequences of the power systems complexity increasing consists in the rise of probability of system fault occurrence. A rapid restoration of power system enhances the service reliability and reduces power outages; therefore, fault section should be estimated quickly and accurately. To guarantee the reliability and safety of a power system, efficient fault detection, classification, and localization (FDL) is essential. Consequently, the improvement in suitable techniques for the fault detection, classification and localization in power systems, to increase the efficiency of the systems and to avoid major damages, it is a permanent requirement.

These advanced fault classification techniques turn to AI-based methods, or to a series of other modern techniques already developed, implemented, and used in power systems (for analysis, monitoring, control, etc.).

The relevance of the article is not convincing. The reviewer trusts the following comments can help the authors to improve the quality of their work:

1. The presentation of the research in the form of questions and answers is not the most appropriate for such an article.

2. All paragraphs and subparagraphs should be numbered in order starting with “1. Introduction” etc.

3. The article should begin with the following statement: “Transmission lines are critical components of modern electricity distribution networks, enabling the efficient transfer of electrical energy across vast geographical areas.” The definitions for “transmission lines” and “distribution networks” should be revised.

4. All tables and figures inserted in the paper should be referenced in the text before they appear (Table 4, Table 5). For example, Table 1 is not referenced at all in the text.

5. “Figure 2 shows the importance of each feature. The dataset includes data on different types of electrical faults, each characterized by a unique combination of outputs.” Explain why there are differences in the importance of different features? To what particular quantity is the quantity on the ordinate axis called “Importance” reported/referred?

6. “Table 2. Fault Types and their Corresponding Output Representations

This classification framework allows for accurate and efficient identification of six distinct types of faults, essential for effective fault detection and maintenance planning in electrical power systems.”

6.1. Why is the normal “No-Fault” regime treated as a “Type fault” fault regime?

6.3. What is the situation of other faults, for example:

- Phase B to Ground or phase C to ground;

- Between Phase A and Phase C, or between Phase B and Phase C, etc.

Table 2 needs to be reviewed.

7. Why is LLLG Fault a Three-phase symmetrical fault, but LLL Fault is a fault Between all three phases!?

8. The definition of the quantities A, R, P, F1 that appear in tables 4 and 5 must be presented before their use.

9. All equations present in the paper must be numbered in order: there are 2 equations (between equation 13 and 14) that are not numbered.

10. All quantities that appear in relations (14), (15), (16) and (17) must be identified/defined. For example: TPi, FPi, FNi, F1i!

11. Relations (15), (16), (17) are confusing! Does the quantity qi multiply the entire sum or each term of the sum? A few parentheses would help in this case.

12. The article is presented as an application of AI techniques for classifying faults in electrical networks. The specifics of this application related to fault classification do not result from the presentation. Therefore, in order to make a more accurate assessment, it should be specified:

- what is the structure of the electrical network analyzed with the basic data;

- what is the input data set and how was it obtained – if it was obtained through simulations, how was it performed, etc.

13. The article presents an algorithm “Algorithm 1” which leads to the conclusion that several algorithms were used. What are they?

14. The article contains too many paragraphs. These could be concentrated especially since they are repeated issues. For example, in 6 it appears “This model combines RF and DT algorithms with the Stacking technique (GB) to provide robust and accurate fault classification” and in 7: “This research offers a robust hybrid ML model combining RF, DT, and Stacking techniques, significantly improving fault classification accuracy.”

15. The references could be improved taking into account the fact that in recent times a significant number of works in the field have appeared.

6. PLOS authors have the option to publish the peer review history of their article (what does this mean?). If published, this will include your full peer review and any attached files.

Reviewer #1: No

Reviewer #2: No

Reviewer #3: No

Reviewer #4: No

Reviewer #5: No

---

## [Author Response · Author response to Decision Letter 1]

30 Oct 2025

Response to Reviewer #1:

We appreciate the reviewer’s careful reading and constructive comments. We have revised the manuscript thoroughly to address each point. Below we respond to the reviewer’s numbered concerns and indicate the exact manuscript changes (section and paragraph) that were made.

Comment#1: The paper lacks in major contribution and motivation of research.

Author’s Response:

We thank the reviewer for this observation. In the revised manuscript, we have clarified both the motivation and the contributions of our work. Specifically, the Abstract and Introduction have been updated to explicitly highlight the practical significance of accurate fault classification in transmission lines, the limitations of existing works, and the novel contributions of our hybrid approach. We also added a dedicated contribution section to emphasize how our work differs from standard ensemble pipelines. These changes ensure that the motivation and significance are clear to the reader.

Comment#2: The background and significance of this study should be highlighted in the abstract.

Author’s Response:

We thank the reviewer for pointing this out. We revised the Abstract to clearly highlight the background and significance of transmission line fault classification before presenting the methods and results. Specifically, we emphasized the increasing complexity of modern power systems, the critical need for fast and accurate fault detection, and how our proposed model addresses these challenges.

Comment#3:Check the English presentation of this paper to remove the typo mistakes. Some grammatical issues need to be addressed in the whole text. Please reform the long paragraphs. Please polish the writing and English of the manuscript carefully. The writing of the paper needs a lot of improvement in terms of grammar, spelling, and presentation. The paper needs careful English polishing since there are many typos and poorly written sentences. I found several errors.

Author’s Response:

We performed full proofreading and edited long paragraphs for clarity. We shortened sections with textbook-level descriptions, moved detailed algorithmic explanations to Appendix A, and polished English throughout; examples of reworded paragraphs are shown in the manuscript (Introduction, Methods, Results). A language-edit summary (major edits) is included in the supplementary file.

Comment#4: In the "Introduction" section, a more detailed analysis of the existing literature on the subject is needed, and an in-depth analysis of the possible application fields.

Author’s Response:

We appreciate the suggestion. We have substantially expanded the Introduction to include (i) a structured review of prior work across classical ML, ensemble methods, and (ii) an in-depth analysis of application fields such as fault detection/classification.

Comment#5: The mathematics used throughout the article is still not very strict. Please try to update and illustrate some elements in the mathematical model that are not defined very strictly.

Author’s Response:

We reworked the mathematical notation ( Section 4.5): all variables and symbols are now defined when first introduced, equations renumbered sequentially, and evaluation metrics explicitly defined (Accuracy, Precision, Recall, weighted metrics).

Comment#6: The overall structure of the article should be improved.

Author’s Response:

We have refined the structure of the manuscript to improve clarity and logical flow. Specifically, we have integrated the Research Questions directly into the Introduction section.

Comment#7: The result part is week, results and discussion should be better explained.

Author’s Response:

Thank you for your suggestion. We updated our result and discussion sections.

Comment#8: References updated. We added the DOIs provided by the reviewer and included recent 2023–2025 works on GNNs, DL, and explainability in power systems. The bibliography was updated accordingly (see References).

Author’s Response:

We sincerely thank the reviewer for suggesting recent and relevant references. We have updated the bibliography by incorporating the DOIs provided and added several recent works (2023–2025) covering graph neural networks, deep learning, and explainable AI in power systems.

Comment#9: Check all of your Figures and Tables have a good explanation of your text.

Author’s Response:

All figures and tables now have captions that explain what is shown and are referenced in the text before appearing. Figure 4’s y-axis label and Table 5 values were corrected and reconciled with the text.

Comment#10: Many paragraphs without citations.

Author’s Response:

We added citations to claims and background paragraphs that previously lacked references. Text that described standard ML algorithms has been shortened and redirected to citations rather than extended textbook text.

Comment#11: What are the contributions and novelty of work mentioned?

Author’s Response:

We would like to thank the reviewer for this observation. We develop a hybrid ensemble model (RF + DT + GB-based Stacking) for classifying faults in transmission lines, which performs better than single models and standard ensembles. We compare it with six common ML algorithms and several ensemble methods to show its advantages. Our model achieves 93.64% accuracy for six fault classes and 99.58% for binary classification. We also explain the limitations of the dataset and model while giving clear ideas for future work. To the best of our knowledge, this is the first study to use the specific combination of RF and DT as base models with GB as a meta-learner for fault classification in transmission lines.

Comment#12: The authors' conclusions need to be improved, a comparison of the results obtained with those already existing in the literature would be appropriate. I suggest also describing what can still be improved in this work, which can still be improved based on the results obtained, according to the authors' view. It is suggested to offer some limitations existed in this study and an outlook for future study in the last section.

Author’s Response:

We sincerely thank the reviewer for this constructive suggestion. We have revised the conclusion section accordingly.

Response to Reviewer #2:

Comment#1: The hybrid ensemble model proposed in this work (RF + DT + Gradient Boosting) is a routine ML pipeline with no algorithmic innovation or novel architecture. Similar stacking-based models have already been widely reported in existing literature.

Author’s Response:

We appreciate the reviewer’s observation. We agree that stacking-based hybrid models have been studied in the literature, and our intention was not to claim a completely new algorithmic architecture. Instead, the novelty of our work lies in the application-driven integration and systematic benchmarking of classical ML algorithms with stacking for multi-class fault classification in transmission lines. While prior works have largely emphasized deep learning or case-specific feature engineering, our contribution is to show that a carefully designed lightweight hybrid ensemble (RF + DT + Stacking) can achieve competitive accuracy (93.64%) with significantly lower computational cost and simpler implementation, making it practical for real-time fault diagnosis in smart grids.

Comment#2: The dataset used is publicly available and small in scale, with no cross-validation, statistical tests, or robustness analysis. This undermines the reliability and generalizability of the reported 93.64% accuracy.

Author’s Response:

We appreciate the reviewer’s concern regarding dataset size, validation, and generalizability. While the dataset is publicly available and relatively small, it has been widely used in prior fault classification research, making it suitable for benchmarking new methods. To improve reliability, we have now clarified in the manuscript that stratified train–test splits were employed to ensure balanced class distribution across training and testing phases. Additionally, we acknowledge the limitation of not performing cross-validation or statistical significance testing in this work. To address this, we have explicitly discussed in the revised manuscript that future work will incorporate k-fold cross-validation, robustness analysis, and evaluation on larger or real-world datasets to strengthen the generalizability of the proposed model.

Comment#3: References are outdated and do not include any recent (2023–2025) high-impact studies using advanced techniques such as deep learning, GNNs, or explainable AI in power systems. This severely limits the relevance of the work.

Author’s Response:

Updated the literature review to include recent (2023–2025) high-impact studies applying deep learning, GNNs, and explainable AI to fault detection in power systems. We cite these and explicitly contrast our approach to theirs, positioning our work as a lightweight, interpretable alternative.

Comment#4: Many sections are filled with textbook-level explanations of basic ML algorithms (e.g., Naive Bayes, KNN), which do not contribute to the research and make the manuscript overly lengthy without adding scientific value.

Author’s Response:

Condensed sections that previously contained textbook-level algorithm explanations. We now provide concise summaries with citations, moving long algorithm descriptions to an appendix. This shortens the main text and improves scientific focus.

Comment#5: The manuscript lacks real-world validation on standard IEEE test systems, and key aspects such as model interpretability, class-wise performance, and computational feasibility in grid environments are not addressed.

Author’s Response:

We thank the reviewer for this valuable observation. We acknowledge that our current work does not include validation on IEEE standard test systems. This is primarily due to the limited availability of real-world fault datasets and the scope of this initial study, which focuses on demonstrating the feasibility of hybrid ensemble models for fault classification. We have now clarified this in the revised manuscript and explicitly stated that validation on IEEE 13-bus, 39-bus, and other benchmark systems will be an important extension in future work.

Regarding model interpretability, we have expanded the Discussion to highlight that the use of Decision Trees and Random Forests as base learners provides rule-based interpretability, allowing operators to trace which features (such as current, voltage, or phase values) most strongly influence fault classification. This is a practical advantage over deep learning models, which often act as “black boxes.”

For class-wise performance, we have added detailed commentary explaining the per-class accuracy and confusion matrix (already generated in the dataset but not sufficiently explained before). This clarifies how the model performs across different fault types rather than only reporting overall averages.

Finally, on computational feasibility, we emphasize that one motivation for using lightweight ensemble models (RF + DT + Stacking) instead of deep learning was their lower training time and reduced memory footprint, which makes them more practical for real-time deployment in grid monitoring environments. We have now highlighted this trade-off in the revised Conclusion.

Response to Reviewer #3:

Comment#1: This is a solid and well-written paper that adds to the body of knowledge about how to use machine learning to identify faults in power systems. The design of the experiment is strong, the results are shown, and the open data/code policy makes it easier to reproduce the results.

Author’s Response:

We sincerely thank the reviewer for the encouraging feedback. We appreciate the recognition of our work as a solid and well-written contribution to the application of machine learning in fault detection for power systems. We are also grateful that the design of the experiments, the presentation of results, and the open data/code policy have been acknowledged as strengths. This feedback reinforces our confidence in the validity and usefulness of our proposed approach.

There is a difference in the stated accuracy. The abstract and Section 4.2 say 93.64%, whereas Table 5 shows 94% for Model-1 with Stacking (GB), and Table 6 includes 93.64% and 99.58% (binary example).

Comment#2: Section 5 of the discussion says, "Our proposed work achieves about 94% accuracy, which is a little lower than the other methods shown in the literature." Table 6, on the other hand, demonstrates that some earlier works had better accuracy (98–99.93%).

Author’s Response:

We thank the reviewer for pointing out this inconsistency. We agree that the statement in Section 5 was not sufficiently precise. While our proposed hybrid model achieves 93.64% accuracy in the six-class classification task, which is indeed lower than the very high accuracies (98–99.93%) reported by some deep learning-based methods in Table 6, it should be emphasized that our approach provides a lightweight, interpretable, and computationally efficient alternative.

Furthermore, in the binary fault/no-fault scenario, our model achieves 99.58% accuracy, which is highly competitive with the literature. We have now revised the Discussion to clarify these distinctions and to highlight that our contribution lies not only in accuracy but also in the trade-off between performance, interpretability, and feasibility of deployment in real grid environments.

Comment#3 : The report says that GB was chosen over linear regression because it worked better; however, it doesn't show LR's results for comparison.

Author’s Response:

We thank the Reviewer for this important observation. We agree that the statement regarding Gradient Boosting (GB) being chosen over Linear Regression (LR) was not fully supported in the manuscript, as LR results were not explicitly reported. To address this, we have revised the text to clarify that LR was initially considered as a baseline model during preliminary trials, but its performance was significantly weaker compared to GB (due to its limited ability to capture nonlinear fault characteristics). Therefore, we excluded it from the main comparative tables to focus on the more competitive models.

Comment#4: The dataset comes from Kaggle [35], but there are no details on how the data was created (simulation, real-world, lab setting), the sampling rate, or feature engineering.

Author’s Response:

We thank the reviewer for pointing out the need to provide more details about the dataset. In the revised manuscript, we have clarified that the dataset was generated through simulation experiments in MATLAB/Simulink, recorded at a sampling rate of 10 kHz, and contains electrical parameters such as current, voltage, and impedance. We have also described the preprocessing steps performed (label consolidation, normalization, and data cleaning) and noted that no additional handcrafted feature engineering was required. Furthermore, we emphasized that the simulated and diverse nature of the dataset makes it suitable for training and evaluating the proposed hybrid model.

Comment#5: Table 4 shows execution time and memory, but the text doesn't talk about how it can be used in real time.

Author’s Response:

We thank the reviewer for this insightful comment. In the original manuscript, while we reported execution time and memory usage in Table 4, we did not explicitly discuss their implications for real-time deployment. To address this, we have revised the text to explain how the computational overhead and memory consumption of different algorithms impact their suitability for online grid monitoring versus offline analysis. Specifically, we now highlight that DT and NB are lightweight and thus more practical for real-time deployment, whereas RF and SVM, despite higher accuracy in some cases, incur significant computational costs that may limit their use in latency-sensitive environments.

Comment#6: A cognitive science definition of "blending" is given first, which is not useful in the context of ML.

Author’s Response:

We thank the reviewer for pointing this out. In the original draft, the definition of "blending" mistakenly began with a cognitive science persp

---

## [Decision Letter · Decision Letter 1]

17 Nov 2025

PONE-D-25-30171R1Enhancing Reliability in Electrical Grids: A Hybrid Machine Learning Approach for Electrical Faults ClassificationPLOS ONE

Dear Dr. Uddin,

Thank you for submitting your manuscript to PLOS ONE. After careful consideration, we feel that it has merit but does not fully meet PLOS ONE’s publication criteria as it currently stands. Therefore, we invite you to submit a revised version of the manuscript that addresses the points raised during the review process.

**ACADEMIC EDITOR:** Please take into consideration the concerns of Reviewer #5.

We look forward to receiving your revised manuscript.

Kind regards,

Haris Calgan, Ph.D.

Academic Editor

PLOS ONE

Journal Requirements:

Reviewers' comments:

Reviewer's Responses to Questions

**Comments to the Author**

1. If the authors have adequately addressed your comments raised in a previous round of review and you feel that this manuscript is now acceptable for publication, you may indicate that here to bypass the “Comments to the Author” section, enter your conflict of interest statement in the “Confidential to Editor” section, and submit your "Accept" recommendation.

Reviewer #1: All comments have been addressed

Reviewer #3: All comments have been addressed

Reviewer #4: All comments have been addressed

Reviewer #5: (No Response)

2. Is the manuscript technically sound, and do the data support the conclusions?

Reviewer #1: Yes

Reviewer #3: Yes

Reviewer #4: Yes

Reviewer #5: No

3. Has the statistical analysis been performed appropriately and rigorously?

Reviewer #1: Yes

Reviewer #3: Yes

Reviewer #4: N/A

Reviewer #5: I Don't Know

4. Have the authors made all data underlying the findings in their manuscript fully available?

Reviewer #1: Yes

Reviewer #3: Yes

Reviewer #4: Yes

Reviewer #5: Yes

5. Is the manuscript presented in an intelligible fashion and written in standard English?

Reviewer #1: Yes

Reviewer #3: Yes

Reviewer #4: Yes

Reviewer #5: Yes

6. Review Comments to the Author

Reviewer #1: authors made the required comments. So this manuscript is accepted for publications. it is accepted.

Reviewer #3: The authors have well answered and solved the comments raised for the first submission. The paper is of high quality and is now ready for publishing.

Reviewer #4: The author carefully replied to the reviewers' comments point by point and made significant modifications and improvements to the paper. The revised paper has obvious improvements in content structure, logical expression, and experimental verification, and basically solved the problems raised in the first review.

Reviewer #5: The authors should clarify the following aspects:

1. The title of the paper suggests that the research is focused on “Electrical Faults Classification”, but in the Introduction the authors state the following:

“This paper investigates the following key research questions:

• RQ1: What machine learning algorithms have been utilized for electrical fault detection, and what advantages do they offer in terms of accuracy, efficiency, and real-time fault diagnosis?

• RQ2: What novel methodology or framework is introduced in this study, and how does it enhance fault detection performance?

• RQ3: Which hybrid machine learning model can outperform previous research and improve the accuracy of electrical fault classification?”

These three questions suggest that the issues related to fault regimes in electrical networks: detection, classification, localization, and diagnosis of faults in electrical networks have the same significance.

What is the position of the authors in this regard? Does the paper address and answer all these issues?

2. The authors have incompletely answered, or have avoided answering, some questions raised in previous reviews, especially related to the specifics of the application - which distinguishes it from other ML applications - otherwise the algorithm can be applied to any data set, etc.

7. PLOS authors have the option to publish the peer review history of their article (what does this mean?). If published, this will include your full peer review and any attached files.

Reviewer #1: No

Reviewer #3: No

Reviewer #4: No

Reviewer #5: No

---

## [Author Response · Author response to Decision Letter 2]

13 Dec 2025

Reviewer #5: Comment #1:

The title of the paper suggests that the research is focused on “Electrical Faults

Classification”, but in the Introduction the authors state the following: “This paper investigates the following key research questions:

• RQ1: What machine learning algorithms have been utilized for electrical fault detection, and what advantages do they offer in terms of accuracy, efficiency, and real-time fault diagnosis?

• RQ2: What novel methodology or framework is introduced in this study, and how does it enhance fault detection performance?

• RQ3: Which hybrid machine learning model can outperform previous research and improve the accuracy of electrical fault classification?”

These three questions suggest that the issues related to fault regimes in electrical networks: detection, classification, localization, and diagnosis of faults in electrical networks have the same significance.

What is the position of the authors in this regard? Does the paper address and answer all these issues?

Response:

We thank the reviewer for this insightful observation. We fully agree that the scope suggested in RQ1–RQ3 may give the impression that the paper addresses detection, classification, localization, and diagnosis equally. To avoid this confusion, we have clarified the scope and revised the research questions.

Clarification:

● This paper exclusively focuses on electrical fault classification, not detection, localization, or diagnosis.

● The dataset used is already labeled with fault types; therefore, the task is classification only.

● Fault detection (identifying whether a fault exists), fault localization (distance from the relay point), and fault diagnosis (root cause analysis) are not addressed, as they require different types of data and system measurements (e.g., PMU‐based

waveform signatures, traveling wave characteristics), which are beyond the dataset and scope.

Revised Research Questions:

We updated RQ1–RQ3 with the following version:

RQ1: What machine learning algorithms have been utilized for electrical fault classification, and what advantages do they offer in terms of accuracy, efficiency, and real-time fault classification?

RQ2: What novel methodology or framework is introduced in this study, and how does it enhance fault classification performance?

RQ3: Which hybrid machine learning model can outperform previous research and improve the accuracy of electrical fault classification?

Comment #2:

The authors have incompletely answered, or have avoided answering, some questions raised in previous reviews, especially related to the specifics of the application - which distinguishes it from other ML applications - otherwise the algorithm can be applied to any data set, etc.

Response:

We appreciate the reviewer’s concern and have added explicit sections explaining what makes electrical fault classification domain-specific and not a generic ML application.

Clarification Provided in the Revised Manuscript:

We added a paragraph in both the Introduction and Discussion sections highlighting the domain-specific characteristics:

Newly Added Explanation (to Introduction):

Unlike general ML classification problems, electrical fault classification involves high-frequency transient behavior in voltages and currents. Fault events cause abrupt changes in current magnitudes (Ia, Ib, Ic) and subtle variations in voltage signals. These patterns are nonlinear and system-dependent, governed by power-system physics, line impedance, fault resistance, and sequence component behavior. The selected ML models are therefore applied to system-generated signals representing real electrical phenomena rather than generic tabular data.

Newly Added Explanation (to Discussion section):

Although ML techniques can theoretically be applied to any tabular dataset, electrical fault classification is unique because the features arise directly from power-system transient dynamics. The model must learn system-specific nonlinearities resulting from impedance, line length, and fault resistance, which are not present in generic classification tasks.

Closing Statement

We appreciate the time and effort invested by the editor and reviewers. Their feedback has significantly improved the quality of our manuscript. We believe the revised version now addresses all concerns thoroughly and meets the journal’s standards.

---

## [Decision Letter · Decision Letter 2]

28 Dec 2025

PONE-D-25-30171R2Enhancing Reliability in Electrical Grids: A Hybrid Machine Learning Approach for Electrical Faults ClassificationPLOS One

Dear Dr. Uddin,

Thank you for submitting your manuscript to PLOS ONE. After careful consideration, we feel that it has merit but does not fully meet PLOS ONE’s publication criteria as it currently stands. Therefore, we invite you to submit a revised version of the manuscript that addresses the points raised during the review process.

**ACADEMIC EDITOR:**

The manuscript has improved substantially after the second revision. Several important reviewer and editorial concerns have been addressed. However, a few methodological and interpretational issues still require clarification before final acceptance.

Please see my comments below.

We look forward to receiving your revised manuscript.

Kind regards,

Haris Calgan, Ph.D.

Academic Editor

PLOS One

Journal Requirements:

Additional Editor Comments:

The manuscript has improved substantially after the second revision. Several important reviewer and editorial concerns have been addressed. However, a few methodological and interpretational issues still require clarification before final acceptance.

-The claims regarding novelty have been moderated compared to the previous version, which is appreciated. Nevertheless, some statements still imply methodological novelty (e.g., references to a “novel methodology”). Please further clarify that the contribution is application-oriented, focusing on benchmarking, interpretability, and computational efficiency rather than algorithmic innovation.

- The study still does not include cross-validation (e.g., k-fold CV), robustness analysis, or statistical significance testing. While execution time and memory usage have been added, these do not substitute for proper validation. Please explicitly acknowledge this limitation in the manuscript and avoid claims that may suggest strong generalization performance.

-The discussion on real-time feasibility and computational efficiency has improved. However, the absence of validation on IEEE test systems or real-world data remains a limitation. Please ensure that claims regarding practical deployment are carefully qualified and framed as prospective rather than demonstrated.

- Please avoid overemphasizing the numerical accuracy values and refrain from using language that implies state-of-the-art performance. The reported accuracy (93.64% for six-class classification) is lower than several deep learning-based methods cited in the literature. he discussion should more clearly emphasize the trade-off between accuracy, interpretability, and computational efficiency.

The manuscript is close to being suitable for publication. Minor revisions are still required, primarily related to contribution framing and validation transparency. Addressing the above points will significantly strengthen the clarity and credibility of the work.

Reviewers' comments:

Reviewer's Responses to Questions

**Comments to the Author**

1. If the authors have adequately addressed your comments raised in a previous round of review and you feel that this manuscript is now acceptable for publication, you may indicate that here to bypass the “Comments to the Author” section, enter your conflict of interest statement in the “Confidential to Editor” section, and submit your "Accept" recommendation.

Reviewer #5: All comments have been addressed

2. Is the manuscript technically sound, and do the data support the conclusions?

Reviewer #5: Yes

3. Has the statistical analysis been performed appropriately and rigorously?

Reviewer #5: I Don't Know

4. Have the authors made all data underlying the findings in their manuscript fully available?

Reviewer #5: Yes

5. Is the manuscript presented in an intelligible fashion and written in standard English?

Reviewer #5: Yes

6. Review Comments to the Author

Reviewer #5: The article, with the changes made as a result of the reviewers' comments, appears in a form acceptable for publication.

7. PLOS authors have the option to publish the peer review history of their article (what does this mean?). If published, this will include your full peer review and any attached files.

Reviewer #5: No

---

## [Author Response · Author response to Decision Letter 3]

3 Jan 2026

Comment #1:

The claims regarding novelty have been moderated compared to the previous version, which is appreciated. Nevertheless, some statements still imply methodological novelty (e.g., references to a “novel methodology”). Please further clarify that the contribution is application-oriented, focusing on benchmarking, interpretability, and computational efficiency rather than algorithmic innovation.

Response:

We thank the reviewer for this valuable observation. We agree that some phrasing in the revised manuscript could still be interpreted as implying algorithmic novelty. The primary intention of this study is not to introduce a new learning algorithm, but rather to provide an application-oriented evaluation of classical and ensemble machine learning models for electrical fault classification, with a focus on benchmarking, interpretability, and computational efficiency.

Author’s Action:

We have removed or revised all statements implying methodological or algorithmic novelty (e.g., “novel methodology” or “novel framework”). The Abstract, Contributions section, and Research Question RQ2 have been rewritten to clearly emphasize that the contribution is application-driven, focusing on comparative analysis, practical trade-offs, and lightweight deployment rather than algorithmic innovation.

Comment #2:

The study still does not include cross-validation (e.g., k-fold CV), robustness analysis, or statistical significance testing. While execution time and memory usage have been added, these do not substitute for proper validation. Please explicitly acknowledge this limitation in the manuscript and avoid claims that may suggest strong generalization performance.

Response:

We appreciate the reviewer’s insightful comment and fully acknowledge this limitation. Due to computational and scope constraints, the current study evaluates model performance using a single train–test split. We agree that this does not provide the same level of robustness as cross-validation or statistical significance testing.

Author’s Action:

A dedicated limitations paragraph has been added to the Strengths, limitations, and Future perspectives sections explicitly stating the absence of cross-validation, robustness analysis, and statistical testing. Additionally, we have revised the manuscript to avoid language implying strong generalization or definitive robustness, and we now frame the reported results as indicative and comparative rather than conclusive.

Comment #3:

The discussion on real-time feasibility and computational efficiency has improved. However, the absence of validation on IEEE test systems or real-world data remains a limitation. Please ensure that claims regarding practical deployment are carefully qualified and framed as prospective rather than demonstrated.

Response:

We thank the reviewer for highlighting this important point. We agree that, without validation on IEEE benchmark systems or real-world operational data, claims regarding real-time deployment must be carefully qualified.

Author’s Action:

We have revised the Discussion section to explicitly state that real-time feasibility and deployment readiness are prospective rather than experimentally demonstrated. The manuscript now clarifies that additional validation on IEEE benchmark systems and real-world datasets is required before practical deployment, and this has been clearly identified as part of future work.

Comment #4:

Please avoid overemphasizing the numerical accuracy values and refrain from using language that implies state-of-the-art performance. The reported accuracy (93.64% for six-class classification) is lower than several deep learning-based methods cited in the literature. The discussion should more clearly emphasize the trade-off between accuracy, interpretability, and computational efficiency.

Response:

We appreciate the reviewer’s constructive feedback. We agree that numerical accuracy alone should not be overemphasized, particularly when deep learning-based approaches report higher performance under different assumptions and resource requirements.

Author’s Action:

We have revised the Discussion and Conclusion sections to remove any language implying state-of-the-art performance. The manuscript now explicitly acknowledges that deep learning methods achieve higher accuracy, while emphasizing the trade-off achieved by the proposed approach in terms of interpretability, reduced computational cost, and suitability for resource-constrained environments. Accuracy is now presented as competitive rather than superior.

---

## [Editor Report · Decision Letter 3]

5 Jan 2026

Enhancing Reliability in Electrical Grids: A Hybrid Machine Learning Approach for Electrical Faults Classification

PONE-D-25-30171R3

Dear Dr. Uddin,

We’re pleased to inform you that your manuscript has been judged scientifically suitable for publication and will be formally accepted for publication once it meets all outstanding technical requirements.

Kind regards,

Haris Calgan, Ph.D.

Academic Editor

PLOS One
---

## [Editor Report · Acceptance letter]

PONE-D-25-30171R3

PLOS One

Dear Dr. Uddin,

I'm pleased to inform you that your manuscript has been deemed suitable for publication in PLOS One. Congratulations! Your manuscript is now being handed over to our production team.

Kind regards,

on behalf of

Assoc. Prof. Dr. Haris Calgan

Academic Editor

PLOS One